# Surface Modification of the Ti-6Al-4V Alloy by Anodic Oxidation and Its Effect on Osteoarticular Cell Proliferation

Itzel P. Torres-Avila [1], Itzia I. Padilla-Martínez [1], Nury Pérez-Hernández [2],
Angel E. Bañuelos-Hernández [3], Julio C. Velázquez [4], José L. Castrejón-Flores [1,*]
and Enrique Hernández-Sánchez [1,*]

[1] Instituto Politécnico Nacional, UPIBI, Avenida Acueducto s/n, Barrio La Laguna Ticomán, Mexico City 07340, Mexico; itzelpam_9318@hotmail.com (I.P.T.-A.); ipadillamar@ipn.mx (I.I.P.-M.)

[2] Instituto Politécnico Nacional, ENMH, Laboratorio de Microbiología, Av. Guillermo Massieu Helguera 239, La Escalera, Mexico City 07320, Mexico; nperezh@ipn.mx

[3] Programa de Posgrado en Farmacología, Centro de Investigación y de Estudios Avanzados del Instituto Politécnico Nacional, Ciudad de Mexico 07360, Mexico; abanuelosh@gmail.com

[4] Departamento de Ingeniería Química Industrial ESIQIE, Instituto Politécnico Nacional, UPALM, Edificio 7, Zacatenco, México City 07738, Mexico; drjcvamx80@gmail.com

* Correspondence: jlcastrejon@ipn.mx (J.L.C.-F); enriquehs266@yahoo.com.mx (E.H.-S.);
  Tel.: +52-1-554-443-3569 (J.L.C.-F.); +51-1-551-069-2992 (E.H.-S.)

**Abstract:** This investigation describes the formation of crystalline nanotubes of titanium oxide on the surface of a Ti-6Al-4V alloy and its biological evaluation. The formation of nanotubes was performed by the anodic oxidation technique with a constant work potential of 60 V but with different anodizing times of 10, 20, 30, 40, 50, and 60 min used to evaluate their effects on the characteristics of the nanotubes and their biological activity. A mixture of ethylene glycol, water, and ammonium fluoride ($NH_4F$) was used as the electrolytic fluid. Scanning electron microscopy (SEM) and X-ray diffraction (XRD) were applied to determine the morphology and crystalline nature of the nanotubes, showing a well-defined matrix of nanotubes of titanium oxide with a crystalline structure and a diameter in the range of $52.5 \pm 5.13$ to $95 \pm 11.92$ nm. In contrast, the XRD patterns showed an increase of defined peaks that directly correlated with treatment times. Moreover, in vitro assays using an innovative cell culture device demonstrated that the inner diameter of the nanotubes directly correlated with cell proliferation.

**Keywords:** anodic oxidation; surface modification; titanium nanotubes; crystalline structure

## 1. Introduction

Titanium and its alloys form a passivating surface film of $TiO_2$ when they come into contact with the surrounding oxygen [1,2]. This surface film gives titanium its high resistance to corrosion by avoiding direct interactions between metal and the environment, turning it into a biocompatible material [3]. However, this feature, which makes titanium highly attractive for some applications, might have a direct impact on cells by inhibiting cellular adhesion, hindering the growth of surrounding tissue, and eventually causing the loosening and loss of prostheses [4]. Different techniques have been applied to modify titanium's surface to improve its cellular response. Microtopography has been found to promote mechanisms such as anchoring and osteoblast functions by facilitating better contact with the bone [5,6]. Nevertheless, studies have shown that microtopography reduces osteoblast proliferation, resulting in a lower mass of bone tissue [7]. On the other hand, advances

in nanotechnology sciences have allowed the generation of nanostructured surfaces that improve the properties and biocompatibility of materials. These surface modifications can be achieved using different methodologies, including chemical vapor deposition (CVD), physical vapor deposition (PVD), and thermochemical processes (boriding, nitriding, and carburizing). Additionally, electrochemical methods, primarily anodic oxidation (AO), are a convenient and inexpensive way to produce $TiO_2$ nanotubes (TNTs), which are highly effective when used in tissue engineering [8]. AO allows one to erode the surface of metals through an electrochemical process via reactions between electrodes driven by an electric field [9]. The diffusion of oxygen ions leads to the formation of an oxide film on the surface of the anode. The morphology, chemical composition, and thickness of the oxide layers can be modified depending on the parameters of the anodizing process. The settings that determine the characteristics of the resulting coatings are mainly the used electrolyte (organic or aqueous) and its composition, pH, treatment time, and applied voltage [10]. Depending on these variables, two types of layers can be obtained. The first is a compact layer known as the barrier layer, which is formed from pore-free titanium oxide, while the second is a duplex layer with a porous or tubular layer formed at the outer part of the film [8]. The formation of a porous or tubular film depends on whether or not inter-tubular spaces have developed between the pores and the barrier layer of the metal-oxide interface [11].

The surface modification of titanium aims to enhance the metal's biocompatibility by modifying the roughness, wettability, and electrical charge of the surface. At the same time, the inner diameter of the nanotubes formed is directly correlated with biocompatibility [12,13]. Pioneering studies in nano surface modification demonstrated that nanotubes with diameters of 100 nm did not affect cell growth in immortalized mouse embryonic cells but did induce cell proliferation. Additionally, alkaline phosphatase activity was shown to be an essential marker for osteoblast differentiation, although other studies have shown opposite results, demonstrating that an inner diameter larger than 55 nm resulted in a dramatic reduction in cell viability [6]. Recent studies have shown that a maximum inner nanotube diameter of 66 nm titanium seems to have a significant effect in the expression of adhesive molecules because shorter diameters did not impact cellular adhesion and growth [14]. On the other hand, Park et al. found that cell adhesion and spreading were severely impaired on nanotube layers with a tube diameter larger than 50 nm, reporting that nanotubes with an inner diameter between 30 and 50 nm are a critical borderline for cell fate [15]. Ming-Ying et al. reported that fibroblast cells exhibited evident diameter-dependent behavior on both as-grown and Ag-decorated $TiO_2$ nanotubes. The authors mentioned that 25-nm-diameter Ag-decorated nanotubes exhibited the most biological activity in promoting the adhesion and proliferation of human fibroblasts [16].

Additionally, different nanoporous derived structures have been generated, including cerium-containing or nano-silver/graphene titanium oxide nanotubes, to provide other important biological activities [17].

Most biocompatibility in vitro assays use a piece of the testing material immersed in the medium culture in conventional culture plates to mimic the interactions of the cells with the material [18–20]. Although this is a widely used configuration, it might not truly represent the complexity of the interactions between the cells and the biomaterial. Thus, the development of a culture device that not only guarantees the complete interaction of the tested material with the cells at all times but also allows different in vitro assays to be performed is necessary.

Thus, this research is focused on the development of an innovative cell culture device named a direct cell culture device (DCD) (patent registration number: MX/a/2019/006316). The DCD was intended to be a conventional culture plate but can be constructed with the material to be tested. The DCD is integrated for three compounds: (a) a base where the cells are placed to be cultured, (b) a barrier to contain the cell and culture fluids, and (c) an inert seal to retain the culture fluids inside the device. The cells are cultured on the base of the DCD, and then (after the required time), the DCD is dismounted, and the base can be set inside an electron microscope or any other equipment for the material–cell interaction analysis. Additionally, DCDs manufactured with the Ti-6Al-4V alloy

were modified by anodic oxidation and were characterized by scanning electron microscopy (SEM), X-ray diffraction, and wettability. Finally, the biocompatibility of the generated TNTs was tested using osteoarticular cell lines in an innovative cell culture device (DCD) [21].

## 2. Materials and Methods

### 2.1. Anodizing Treatment

Cylindrical samples of Ti-6Al-4V biomedical alloy (ASTMB348 Gr5, Química Islas, Mexico City, Mexico), the chemical composition of which is provided in Table 1, was used as the substrate in this study.

**Table 1.** Chemical composition of Ti-6Al-4V alloy (wt.%)

| C | Fe (Max) | H (Max) | N (Max) | O (Max) | V | Al | Ti |
|------|----------|---------|---------|---------|---------|-----------|---------|
| 0.08 | 0.30 | 0.015 | 0.05 | 0.2 | 3.5–4.5 | 5.5–6.75 | Balance |

Samples were sectioned with 19 mm in diameter and 4 mm in depth and were prepared by a conventional metallographic process to obtain a mirror-surface finished. Then, the samples were cleaned in an ultrasonic bath for 5 min in ethanol and deionized water. After washing, samples were mounted in an electrochemical cell, as shown in Figure 1.

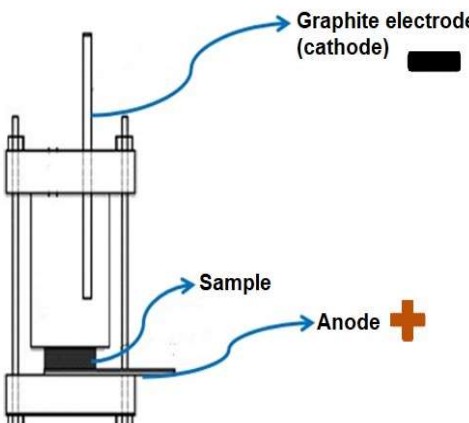

**Figure 1.** Schematic representation of the electrochemical cell for nanostructure modification.

The samples were used as the anode, while an electrode of graphite was used as a counter electrode separating them with a constant distance of 10 mm. The anodization process was carried out using a mixture of ethylene glycol (EG, 99.8% purity), 1 wt.% of distilled water, and 0.5 wt.% of $NH_4F$ (99.98% purity) (conductivity 1.23 mS cm$^{-1}$ and pH 7.4), were purchased from Sigma-Aldrich (Toluca, Mexico) and used as supplied. The work potential was established in 60 V, and the exposure time was set at 10, 20, 30, 40, 50, and 60 min. After the anodic oxidation treatment, the samples were rinsed with deionized water and dried in hot air.

### 2.2. Surface Characterization

The nature of the modified surfaces was examined using an Olympus GX51 optical microscope (Olympus, Center Valley, PA, USA) and a scanning electron microscope (SEM) (JSM-6360LV, JEOL, JEOL Ltd., Akishima, Japan).

Additionally, the structures of the modified surfaces were analyzed by the X-ray diffraction technique (XRD) with the aid of a D8 FOCUS diffractometer (Bruker, Billerica, MA, USA), equipped with Cu-Kα radiation (1.5418 Å).

### 2.3. Wettability of the Surfaces

It has been reported that the hydrophilicity of the surfaces plays a crucial role in cell attachment, spread, and cytoskeletal organization. [22]. Thus, it is important to study the wettability of the modified surfaces through the formation of TNTs. The hydrophilicity of the modified surfaces was assessed from the measurements of the contact angle between the deionized water and the sample surface at room temperature using a 20 μL drop volume. Ten different drops for each sample were measured to ensure the reproducibility of the results. The drop images were acquired with a Canon Raptor-3, 16-megapixel digital camera (Tokyo, Japan), and analyzed using an image analyzer.

### 2.4. Design of the Direct Cell Culture Device

A direct cell culture device (DCD) was constructed individually using cylindrical samples of the Ti-6Al-4V biomedical alloy as a cell culture base, according to the process described in the methodology (2.1). Once the DCDs were assembled, they were ultrasonically cleaned to remove any impurities, cleaned with a disinfectant solution, and autoclaved. Then, the DCDs were aseptically manipulated and used for the cell culture (see Figure 2). A total of four independent DCDs were built for each of the conditions evaluated.

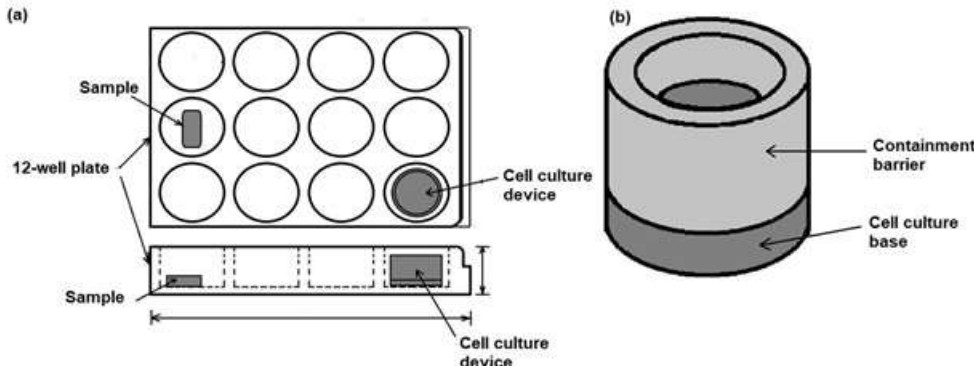

**Figure 2.** Schematic representation of the direct cell culture device (DCD) used for the in vitro experiments. (**a**) Colocation of the DCD for the cell culture and (**b**) configuration of the DCD.

### 2.5. Cell Culture of Osteoarticular Cells

The human fetal osteoblast cell line hFOB 1.19 (ATCC CRL11372) and tumoral chondrosarcoma SW-1353 (ATCC® HTB-94) were acquired from the American Type Culture Collection (Manassas, VA, USA). The cells were grown in Dulbecco's Modified Eagle's Medium (Gibco, Life Technologies, Paisley, UK) and Leibovitz's L-15 Medium (Gibco, Life Technologies, Paisley, UK), respectively, and were supplemented with 10% (*v/v*) fetal bovine serum (FBS JR Scientific, Woodland, CA, USA) and penicillin-streptomycin (Sigma-Aldrich, Toluca, Mexico). The cells were maintained at 37 °C under 5% $CO_2$ in a humidified incubator in the culture media, which was refreshed as needed. The cells were grown in traditional culture plates before being seeded in the DCDs.

The hFOB 1.19 osteoblast cell line was grown at 37 °C to slow the cell growth rate. Our research group tested the differentiation recurrently by monitoring the mineralization with Von Kossa staining. The SV Tag was stable at temperatures below 40 °C, so the current state of our cell line is not differentiated.

### 2.6. Cytotoxicity and Proliferation Assays in Osteoarticular Cells

The direct contact devices were modified to obtain nanostructured Ti-6Al-4V samples, autoclaved (121 °C, 30 min), seeded with the SW-1353 cells (500 μL of a cell suspension of $10^5$ cell/mL), and located in a 12-well polystyrene plate. The cells were cultured for 24 h to allow cell attachment, and the culture medium was then replaced and maintained for 24 h more. Cytotoxicity was measured by lactate dehydrogenase (LDH) released into the culture supernatant or present in the attached lysed

cells as an indirect indicator of cell viability using the CytoTox 96 non-Radioactive Cytotoxicity Assay kit (Promega, Madison, WI, USA). The measurements were performed at 492 nm (wavelength) in a Thermoskan FC device (Thermo Scientific, Waltham, MA, USA). Additionally, proliferation in the DCDs was determined using hFOB 1.19 cells with an MTT in vitro assay. Briefly, hFBO cells were seeded in the DCDs (500 µL of a cell suspension of $10^5$ cell/mL) and placed into a 12-well polystyrene plate for 24 h. Moreover, the selected samples were seeded, and the proliferation was determined on days 3, 7, and 14. After the incubation time finished, 50 µL 3-(4,5-dimethylthiazol-2-yl)-2,5-diphenyltetrazolium bromide (MTT) (Sigma Aldrich, Toluca, Mexico) solution (5 mg/mL in PBS) was added to each well. The samples were then placed in a cell incubator for 4 h, and 500 µL of DMSO was added to the DCDs to solubilize the formazan. The absorbance was read in a multiplate reader at 570 nm. The fold change in activity was measured by dividing the OD of the TNTs over the OD of the non-modified surface.

## 3. Results and Discussions

### 3.1. Surface Characterization

Figure 3 shows SEM images of the untreated sample (Figure 3a) and samples after the anodization process (Figure 3b,c).

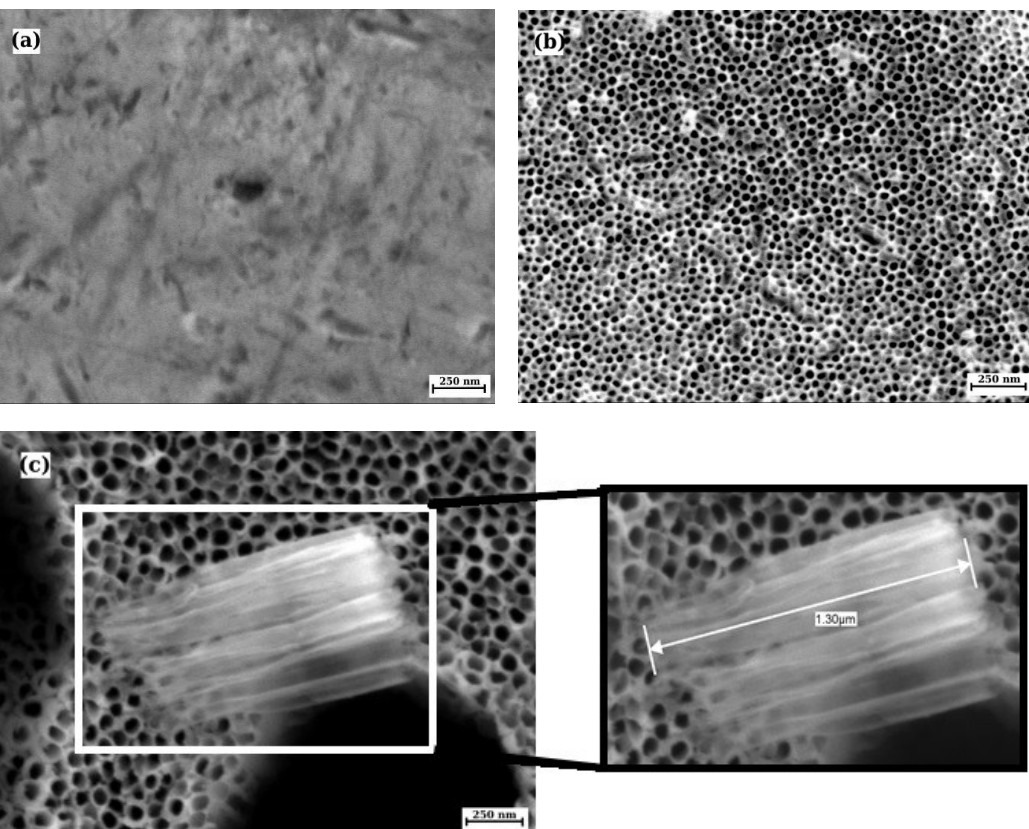

**Figure 3.** SEM images of the untreated surface of Ti-6Al-4V (**a**) and the surface of Ti-6Al-4V anodized for 60 min (**b**,**c**).

The structure of the anodized samples exhibited a homogeneous and uniform array of TNTs compared to the non-treated samples, where an even and smooth surface was observed. The inner diameter of the TNTs ranged from 52.5 ± 5.13 to 95 ± 11.92 nm at a depth of 1.30 ± 0.09 µm on average (Figure 3c). These results suggest that the growth of the TNTs occurred proportionally to the applied potential, with a growth factor $fg \approx 1$–5 nm/V [23]. Importantly, the inner diameter of the TNTs was

determined at the different AO conditions tested, as can be observed in Figure 4, yielding diameters starting at 100 nm, which decreased depending on the treatment time.

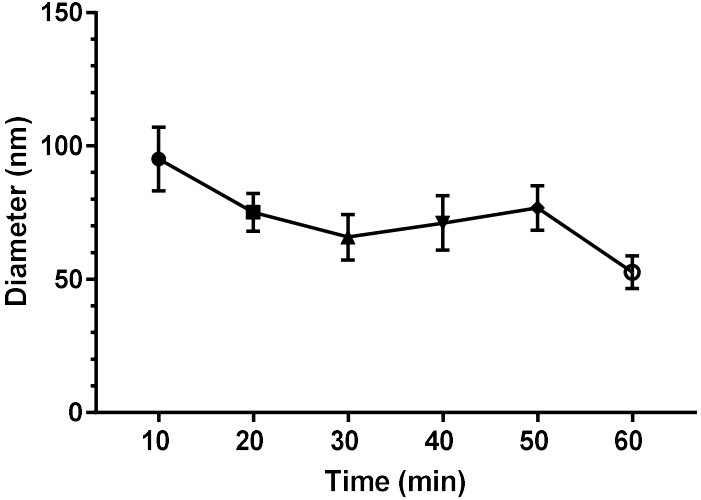

**Figure 4.** Inner diameter of the TNTs as a function of the anodizing time.

The TNTs exhibited a V-shape due to the decrease in their inner diameters from top to bottom along their length (Figure 3c). This condition is not desirable because a V-shape promotes the detachment of the TNTs. Jing Fei Chen et al. [11] reported that it is possible to change a nanotube's profile from a V-shape to a U-shape by increasing the temperature of the electrolyte because this procedure alters the coefficient of diffusion (D) during the anodization process. This increase is proportional to the viscosity of the electrolyte ($1/\eta$), according to the Stokes–Einstein relation [11]. Thus, when the temperature increases, the viscosity of the electrolyte decreases, but D also increases. Consequently, an increase in D leads to an increase in the oxidation process [11,24].

The process of TNT's formation on Ti-6Al-4V titanium alloy surface by AO can be explained by the reactions of oxidation and corrosion as follows:

During the AO, an oxide reaction occurs on the titanium alloy surface where $Ti^{4+}$ ions interact with $OH^-$ and $O^{-2}$ provided by the water in the mixture, as shown in Formulas (1)–(5).

$$Ti \rightarrow Ti^{4+} + 4e^- \tag{1}$$

$$2H_2O + 2e^- \rightarrow 2OH^- + H_2 \tag{2}$$

$$OH^- \rightarrow O^{-2} + H^+ \tag{3}$$

$$H^+ + 2e^- \rightarrow H_2 \tag{4}$$

$$Ti^4 + 2O^{-2} \rightarrow TiO_2 \tag{5}$$

Thus, the formation of titanium oxide occurs as in Formula (6):

$$Ti^{4+} + 4OH^- \rightarrow Ti(OH)_4 \tag{6}$$

In the same way, the anodic layer is hydrated as in (7) to become titanium oxide, as shown in Formula (8). Finally, oxygen is released during formation of the barrier layer as in Equation (9), thereby finishing the reaction process, as described by Formula (10):

$$Ti(OH)_4 \rightarrow TiO_2 + 2H_2O \tag{7}$$

$$2H_2O \rightarrow 4H^+ + 4e^- + O_2 \tag{8}$$

$$2Ti + 6H_2O \rightarrow 2TiO_2 + 6H_2 + O_2 \tag{9}$$

$$Ti(OH)_4 + 6F^- \rightarrow TiF_6^{-2} + 4OH^- \tag{10}$$

Finally, the $F^-$ ions react with $Ti^{4+}$ and the other oxide species to form $TiF_6^{-2}$ (Formula (12)), which is soluble, thereby facilitating the dissolution of the Ti cations:

$$TiO_2 + 6F^- + 4H^+ \rightarrow TiF_6^{-2} + 2H_2O \tag{11}$$

$$Ti^{4+} + 6F^- \rightarrow TiF_6^{-2} \tag{12}$$

It is considered that the presence of $F^-$ ions is crucial in forming porosity in the film because of its capability to dissolve the oxide. Additionally, $F^-$ ions possibly act to sustain the porosity during the formation of the nanotubes. Therefore, the $TiF_6^{-2}$ compounds will remain in the electrolytic solution and should be eliminated by a subsequent cleaning process.

On the other hand, the samples exposed to 10, 20, 30 40, and 50 min of anodization showed an uneven nanostructured film (Figure 5).

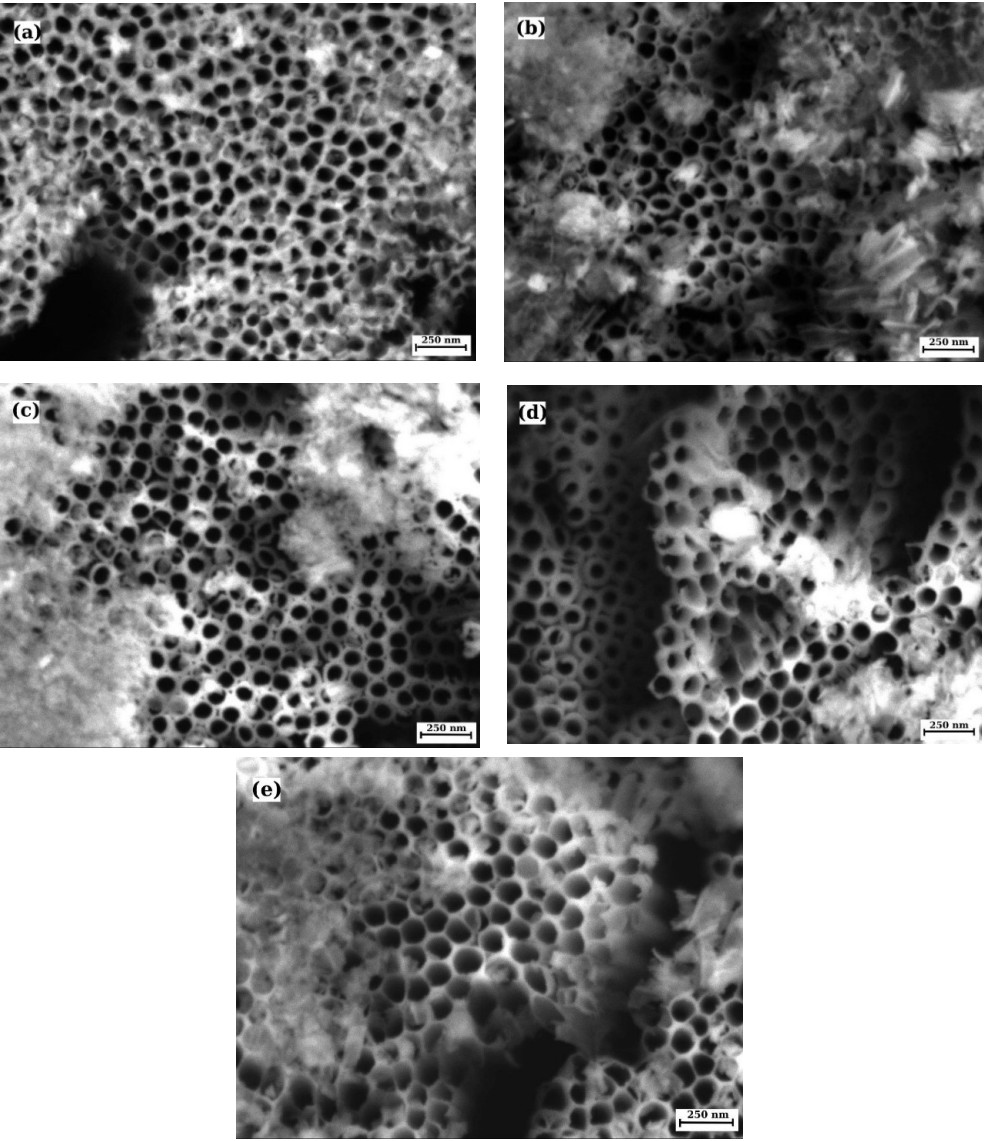

**Figure 5.** SEM images of the anodized surface of Ti6Al4V at different times, 10 min (**a**), 20 min (**b**), 30 min (**c**), 40 min (**d**), and 50 min (**e**).

Additionally, the walls of the TNTs formed on the sample exposed for 10 min (Figure 5a) were not defined. However, the TNT arrays, obtained from 20–40 min of anodization (Figure 5b–d), showed inter-tubular spaces between themselves, with well-defined walls. As the treatment time increased, the TNTs were detached arbitrarily, exposing their shapes.

Moreover, the active area and the morphological quality of the film were reduced as the precipitation of the oxide waste was increased [21]. Nevertheless, the appearance of the sample anodized over 50 min (Figure 5e) was similar to the surface anodized for 10 min, exhibiting a nanoporous surface without inter-tubular spaces. This behavior seen in the TNT film can be explained. As the process evolves, the length of the TNTs increases to a maximum near 1 μm, so when that value is reached, the TNTs tend to detach and a new film of TNTs is re-formed.

The fact that the TNTs reach a limiting value as a function of the anodizing time can be explained by the steady-state when Fluor-containing electrolytes are used. A steady-state occurs when the pore growth rate at the TNT interface reaches the thickness value necessary to reduce the dissolution rate of the oxide film on the external interface. Then, the nanotube oxide layer continuously consumes the titanium substrate without thickening the oxide layer [25].

The nature of the TNTs was evaluated by XRD, as shown in Figure 6.

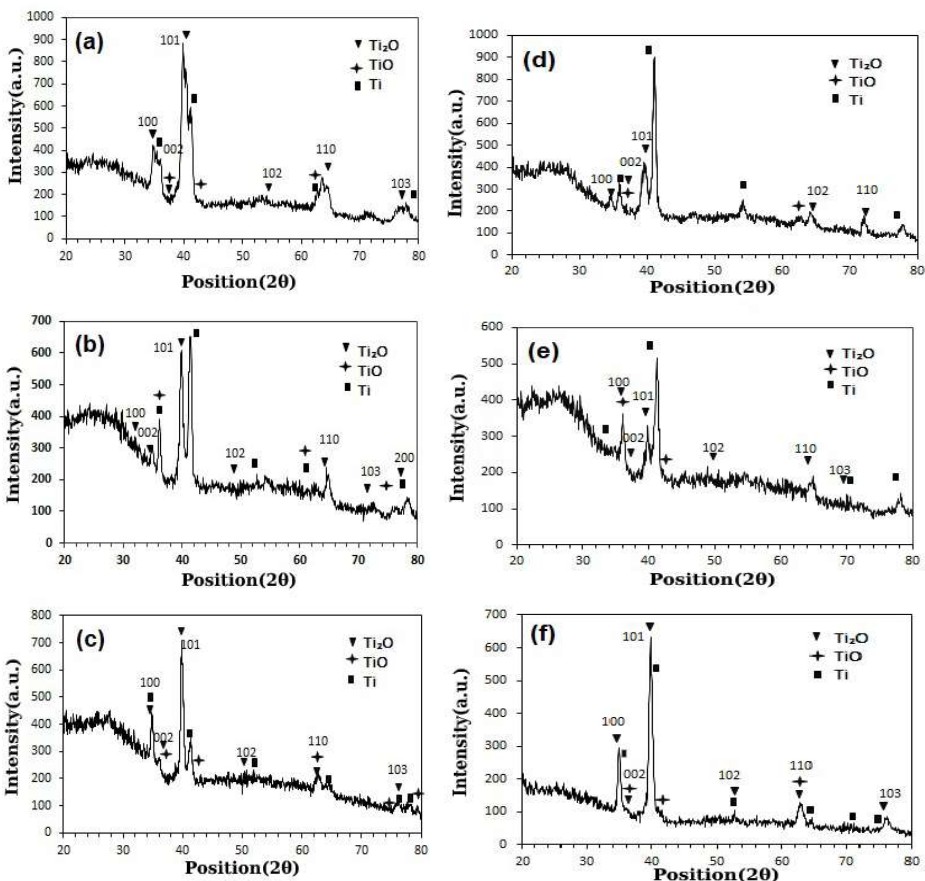

**Figure 6.** XRD pattern of the anodized surface of Ti-6Al-4V at different times, 10 min (**a**), 20 min (**b**), 30 min (**c**), 40 min (**d**), 50 min (**e**), and 60 min (**f**).

Here, a crystalline surface can be observed, where the highest peak indicates the presence of $Ti_2O$. However, this titanium oxide is described as the compact hexagonal arrangement of Ti atoms, where the oxygen atoms occupy every second normal layer of octahedral interstices to the z-axis (parameter c). It is important to note that, according to the results reported in the literature, the TNT crystalline structure seems to be strongly dependent on the electrochemical parameters, such as the applied potential and the anodization time [23]. The structures of the TNTs achieved below 20 V were reported

to be amorphous, while their crystalline structure was reported to appear at higher voltages. Moreover, the crystalline structure was reported to be anatase, rutile, or a mixture of both [10,26,27]. Thus, the higher potential used under these experimental conditions favors the formation of a crystalline structure (Figure 6). This assumption is consistent with that reported by Zixue Su et al. [28], in which crystallization of $TiO_2$ is facilitated by a higher concentration of $NH_4F$, higher voltages, and longer anodization times. Junheng Xing et al. [26] demonstrated that the crystallinity of the TNTs depends strongly on time. According to them, when the anodizing time is short (0.5 h), no Raman peaks can be found for the formed TNTs. In comparison, with the anodizing time of 4 h, four Raman bands, attributed to anatase type of $TiO_2$, can be detected for anodic titanium oxide films. Subsequently, the Raman peaks become sharper and more intensive when the oxidation time was prolonged to 8 and 12 h.

Therefore, as in this work, the $NH_4F$ concentration and the applied voltage remain constant, it is feasible to declare that the crystallinity of the TNTs only depended on the anodization time.

Table 2 and Figure 7 show the Energy Dispersive Spectroscopy assay (EDS) results of the sample anodized for 60 min.

**Table 2.** Chemical composition of the EDS analysis realized to the sample anodized for 60 min.

| C (wt.%) | O (wt.%) | F (wt.%) | Al (wt.%) | V (wt.%) | Ti (wt.%) |
|---|---|---|---|---|---|
| 4.9 | 30.15 | 10.99 | 2.93 | 1.68 | 49.34 |

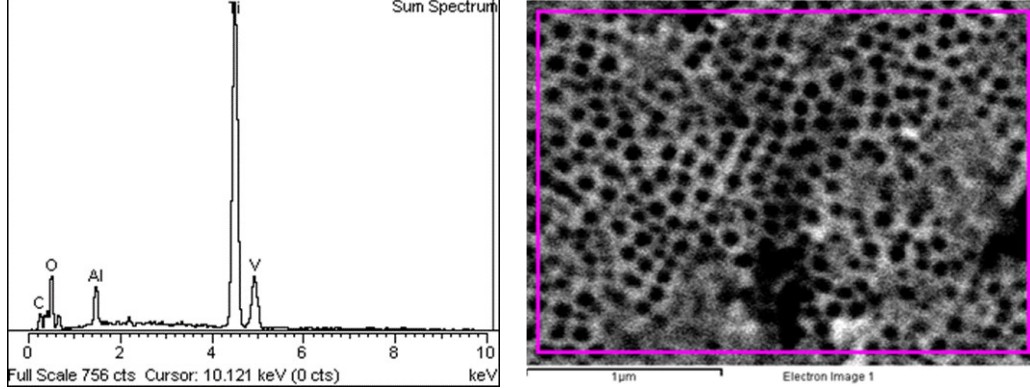

**Figure 7.** EDS patterns realized to the sample anodized for 60 min.

The presence of vanadium (V), carbon (C), and aluminum (Al) indicate the chemical composition of the alloy. Additionally, the increase in the oxygen content corroborates the presence of titanium oxides (and probably V an Al oxides) on the sample surface. Nevertheless, the XRD patterns do not indicate the presence of structures containing the alloying elements, so it is valid to assume that they do not interact with the TNT crystalline structure.

The increase in both carbon and fluor content, probably due to a remnant from the electrolytic solution. Al and V in the anodized surfaces are probably present in the form of $Al_2O_3$ and $V_2O_5$, respectively. $Al_2O_3$ has been used as a highly biocompatible biomaterial. This compound promotes the adhesion and growth of osteoblasts, particularly in its nanostructured form [29]. The effects of $V_2O_5$ are more controversial due to this compound has shown to be genotoxic for lymphocytes and cytotoxic for several fibroblast and tumor cell lines [30,31]. On the other hand, vanadium oxides, including $V_2O_5$, showed antidiabetic effects (by mimicking the effect of insulin), and also induced osteogenic differentiation of mouse embryonic fibroblast cells [32]

Finally, despite the presence of toxic V on the surface of the treated samples (Table 2), the anodizing process caused its oxidation reducing and, subsequently, an evident decrease in the toxic effect compares to the metal substrate (see Table 1).

## 3.2. Wettability of the Surface

The formation of an anodized film at the surface of the Ti-6Al-4V samples influences the hydrophilicity of the treated material, which is an essential factor that affects cellular attachment [1,22]. This surface property is stated by the water contact angle that ranges from 0° on very hydrophilic surfaces to greater than 90° on hydrophobic surfaces. Hydrophilic surfaces exert a higher affinity to proteins than hydrophobic surfaces [33]. In that sense, the hydrophilicity of the samples was estimated by contact angle measurements (Figure 8).

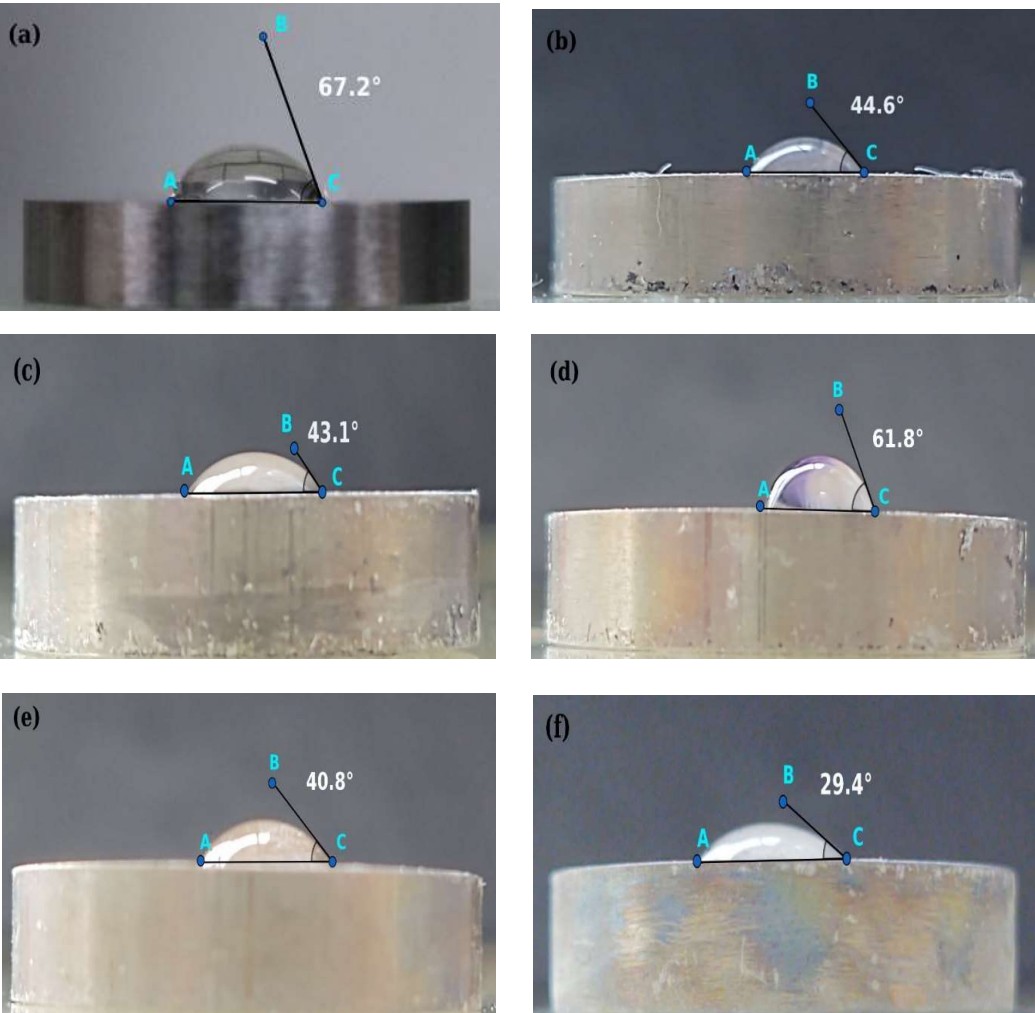

**Figure 8.** *Cont.*

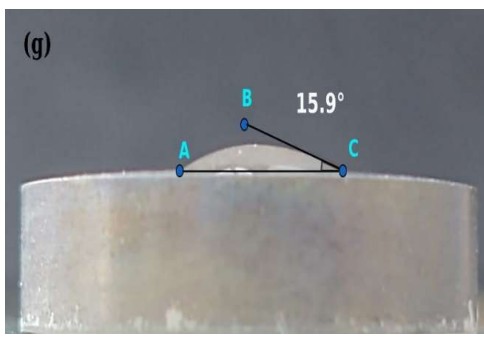
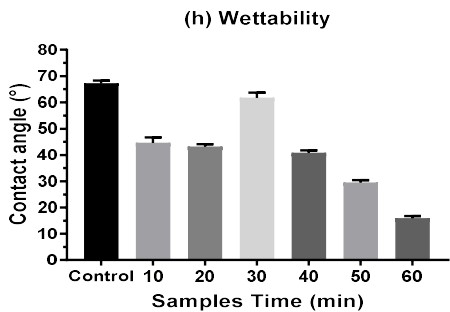

**Figure 8.** Contact angle of (**a**) the non-treated sample and anodized samples for 10, 20, 30, 40, 50, and 60 min, (**b**–**g**), respectively; (**h**) the behavior of the contact angle as a function of the anodization time.

The water contact angles of the seven samples were 67.2° ± 1°, 44.6° ± 2°, 43.1° ± 1°, 61.8° ± 2°, 40.8° ± 1°, 29.4° ± 1, and 15.9° ± 1°. The non-treated sample showed the most extensive contact angle, and the sample anodized over 60 min showed the lowest contact angle. Importantly, an increase in the contact angle was observed in the samples anodized for 30 min. This behavior agrees well with the assumption that the TNTs reach a maximum length value as a function of the anodization time, which was discussed previously.

Furthermore, the contact angle is related to the crystallinity of the TNTs since, as the treatment time increase, the crystallinity of the TNTs increases as well (see Figure 6). The above explains why the non-treated sample exhibited the most hydrophobic surface. Thus, the hydrophobic/hydrophilic relation of the TNTs seems to be related to the amorphous/crystalline, and this behavior could confirm the successful formation of a more bioactive coating on the Ti-6Al-4V surface [2].

This behavior indicates that the initial hydrophobic Ti surface changed to a hydrophilic surface via anodic oxidation, which indicates a significant increase in the capability of the surface to engage in cell proliferation. A possible explanation for the increase in the wettability of the anodized samples was provided by Katić et al. [2]. In their research, the authors conclude that the wetting properties of the modified Ti-implant surface are predominantly affected by the hydrophilic–OH functional groups oriented in the upper part of the coating, possibly originating from glycerol molecules. It is clear that their surface modification technique was different from that used in our study. However, as we used ethylene glycol ($C_2H_6O_2$), it is feasible to assume the presence of OH functional groups on the surface of the TNTs. Additionally, Das et al. reported that a low contact angle indicates high surface energy, which is a key factor contributing to better cell attachment [34].

As a result of anodic oxidation, the hydrophilicity on the modified surface augmented directly correlates with the increase in cellular proliferation compared to the non-treated surface. The effect of hydrophilicity has been described previously, where the incorporation of OH groups on the surface has a direct effect on cellular proliferation and differentiation by increasing the expression of the ALP enzyme [35]. It is possible that the use of ethylene glycol ($C_2H_6O_2$) might result in the addition of OH groups on the surface, as it has been previously reported that the wetting properties of the modified Ti-implant surface are predominantly affected by the hydrophilic–OH functional groups oriented in the upper part of the coating, possibly originating from glycerol molecules.

### 3.3. Cytotoxicity and Proliferation of TNT Determined in the Direct Cell Culture Device Culture

Firstly, the cytotoxic effect was established by the presence of lactate dehydrogenase (LDH) in the supernatants of the SW-1353 cell line cultured in different DCD samples in vitro. When evaluating the samples, there was no presence of the enzyme in the supernatants, demonstrating the absence of cytotoxicity. However, as an indirect measure of viability, the intracellular concentration of LDH was assessed, showing a significant overexpression in the sample anodized over 60 min or with an inner diameter of 52.6 nm (see Figure 9). To validate the findings, the changes in metabolic activity were assessed as a readout of cell growth using the MTT proliferation assay for all the samples with the

hFOB 1.9 cell line, the most biologically relevant cell line. As seen from the LDH assay, the maximal metabolic activity, which indicates viability and proliferation, was observed in the sample anodized for 60 min (see Figure 10).

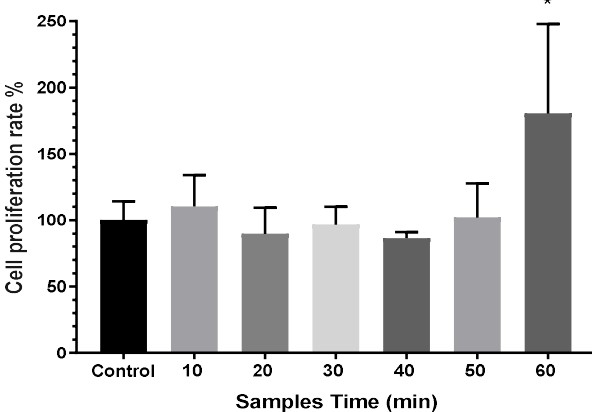

**Figure 9.** SW1353 cell viability determined by the presence of total LDH enzyme at different times. Results are the mean of four independent experiments performed in the DCD ± S.D. A one-way ANOVA was performed between the O.D. of each modified sample and the unmodified sample or control, where (*) ($p < 0.05$).

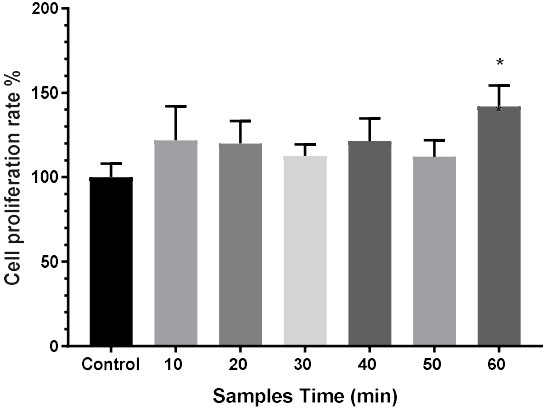

**Figure 10.** hFOB cell proliferation on samples modified at different times, as determined by mitochondrial activity. Results are the mean of four independent experiments performed in the DCD ± S.D. A one-way ANOVA was performed between the O.D. of each modified sample and the unmodified sample or control, where (*) ($p < 0.05$).

The SW1353 cell viability of the sample modified at different times determined by the 60 min TNT was significantly higher than that of the control (Figure 9), and the 60 min TNT produced a considerably higher percentage of cell proliferation than that of the control; (*) One-way ANOVA ($p < 0.05$) (Figure 10).

After demonstrating the effects of the TNT array surface, the viability and proliferation of the human fetal osteoblast cell line hFOB 1.19 was measured on different days using the samples anodized for 10 and 60 min. These samples represent opposite nanotube inner diameter allowing to compare its effect on cellular growth directly. Additionally, an increase in cellular proliferation was seen in the samples treated for 10 and 60 min, where the latest had a statistically significant cell growth when compared with the untreated material. The results show that not only was the viability maintained for every time tested, but a significant threefold change in metabolic activity could also be seen 14 days after the culture. In contrast, most other studies only evaluated proliferation at seven days [14,36,37],

indicating an increase in cell density (see Figure 11). Moreover, other functionalized TNTs did not show an improvement in cell proliferation [17,37].

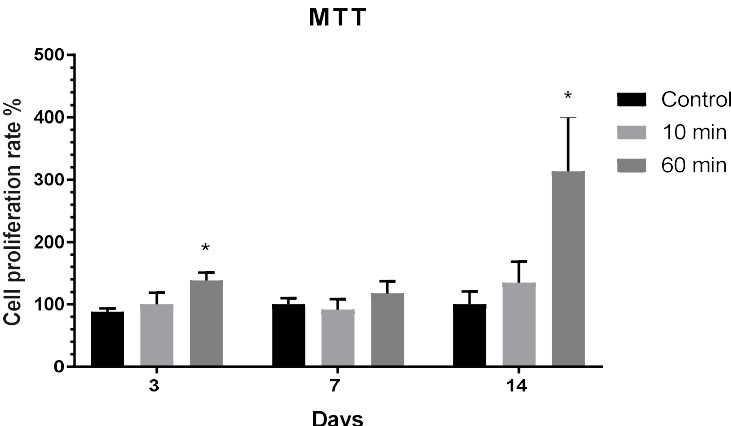

**Figure 11.** hFOB cell proliferation on samples modified at different times determined by mitochondrial activity after several days in a culture. Results are the mean of four independent experiments performed in the DCD ± S.D. A one-way ANOVA was performed between the O.D. of each modified sample and the unmodified sample or control, where (*) ($p < 0.05$).

The increased viability and proliferation seen in the samples anodized for 60 min might be attributed to the differences in the inner diameters of the nanotubes, which seem to affect cell bone proliferation and differentiation. Nanotubes with inner diameters around 60 nm or lower had a positive impact on cell adhesion, proliferation, and differentiation [14]. Importantly, the sample anodized for 60 min, with the best proliferative activity, had an inner diameter below 60 nm. Additionally, although diameters higher than 100 nm support cell proliferation, it has been shown that these diameters might not be ideal [38]. Interestingly, the best TNT diameter is not related to the cell size but the protein size. Oh. et al. [38] reported that proteins as Fibronectin as well as albumin were detected on the surface of TiO$_2$ nanotubes after cultured bovine serum albumin (BSA). As they said, fibronectin and albumin play important roles in cell adhesion, and their size is no mere than several nanometers (≈30–60 nm). Under this context, when TNTs diameter is over 60 nm, the protein aggregated cannot initially stay on the surface of the nanotubes and attach only in the top portion of the walls of the nanotubes. Finally, to know the best nanotube diameter is crucial in the initial protein adhesion, which determines the degree of cell adhesion.

On the other hand, although the roughness of the surface of the nanotubes is considered as an essential parameter for cell interaction, from the nanoscale perspective, the nanotubes are generally of uniform height and diameter. Of course, there are slight variations. Studies have shown that differences in the processed sample surface range from 200 nm to 20 μm [38]. These values of roughness may be considered irrelevant from the context of what a cell sees.

### 3.4. Osteoblast Morphology at Different Times

The samples anodized for 60 min were incubated with the hFOB 1.9 cell line on different days in the DCD, and SEM images were obtained directly to determine the cell morphology and density (see Figure 12).

After three days of culture, numerous cells were scattered and attached to the TNT surface (Figure 12a). A magnified image showed that the cells had a rounded but not a triangular or polygonal shape, as reported previously [14,37], which suggests proliferation according to the increased metabolic and LDH enzymatic activity (Figure 12b). Importantly, the cells were attached to the surface of the nanopore with interactions among them. Then, the samples were analyzed after 14 days of culture in the DCD. A scattered pattern was not evident, although the continuous layer covering the surface of

the TNTs was observed (Figure 12c). A magnified image of the same sample showed that, similar to day 3, the cells were attached to the surface with interactions among them (Figure 12d). Although our results showed not only changes in viability but also an increase in cell proliferation, the cells, at least morphologically, did not show a differentiated phenotype. Nonetheless, several differentiation markers, including vinculin, collagen type 1, alkaline phosphatase, and osteocalcin, need to be evaluated [37].

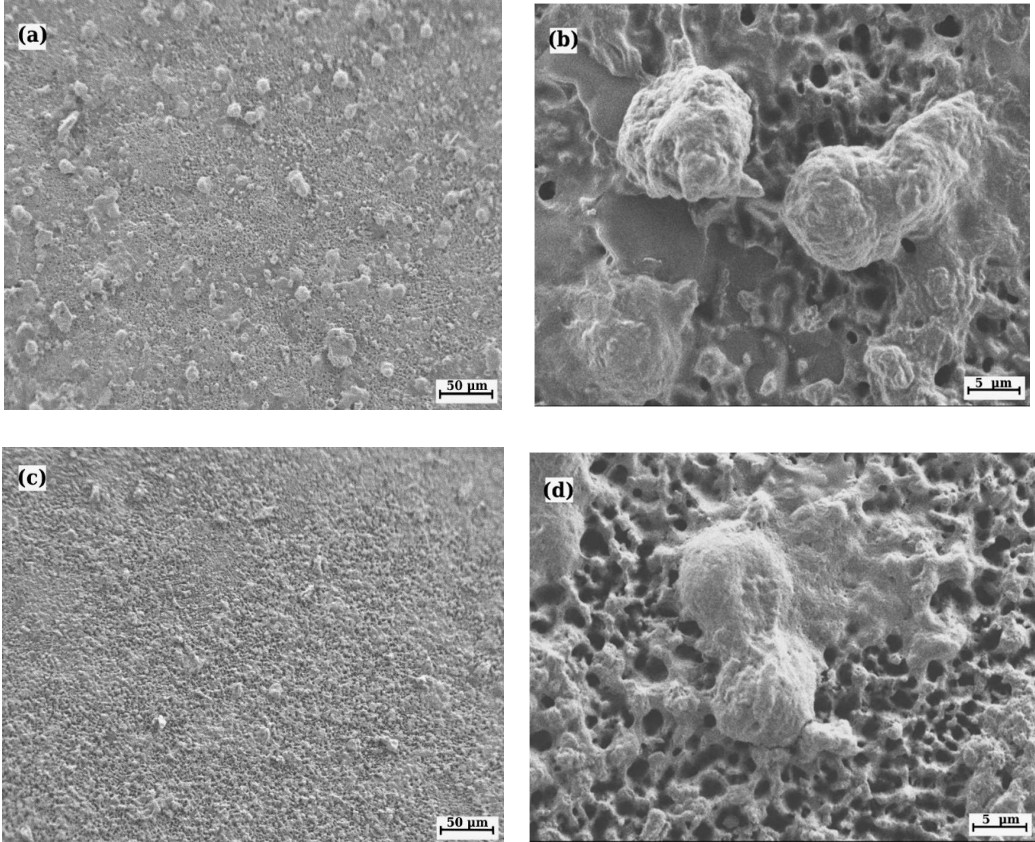

**Figure 12.** SEM images of osteoblast hFOB for the sample anodized for 60 min after three days (**a**,**b**) and after 14 days of culture (**c**,**d**).

## 4. Conclusions

The conclusions of this study can be summarized as follows:

The anodic oxidation of the Ti-6Al-4V alloy was realized in an electrolyte base ethylene glycol with 0.5 wt.% of $NH_4F$ and 1 wt.% of water and a potential difference of 60 V.

The effect of anodization time was visible in the formation of the nanotube matrix, especially in the inner diameter of the nanotubes. The samples anodized for 60 min showed greater uniformity than those exposed to a shorter anodization time.

In contrast, the inner diameter was reduced at longer anodization times. On the other hand, according to the XRD patterns, the obtained nanotubes were crystalline titanium oxide ($Ti_2O$), which not only have not been reported before but have also not been evaluated biologically. Moreover, the in vitro experiments using modified DCDs demonstrated that viability was not only maintained after the de-anodizing process but also that the nanotubes with inner diameter of 60 nm promoted cell proliferation. Finally, the SEM images showed that the cells were attached to the surface with interactions between them.

## 5. Patents

As a result of the work reported in this manuscript, the patent registration with the number: MX/a/2019/006316, referred to as the "direct cell culture device" (DCD) was generated, which is mentioned in Section 2.4.

**Author Contributions:** Conceptualization: N.P.-H., J.L.C.-F., and E.H.-S.; Data curation: I.P.T.-A. and A.E.B.-H.; Formal analysis: N.P.-H., A.E.B.-H., and E.H.-S.; Funding acquisition: N.P.-H., J.L.C.-F., and E.H.-S.; Investigation: I.P.T.-A., I.I.P.-M., and J.C.V.; Methodology: I.P.T.-A., A.E.B.-H., and J.C.V.; Project administration: N.P.-H., J.L.C.-F., and E.H.-S.; Supervision: I.I.P.-M., A.E.B.-H., and J.C.V.; Validation: I.P.T.-A. and N.P.-H.; Visualization: N.P.-H. and E.H.-S.; Writing—original draft: I.P.T.-A. and N.P.-H.; Writing—review and editing: J.L.C.-F. and E.H.-S. All authors have read and agreed to the published version of the manuscript.

**Funding:** This work was supported by research Grant 2101 of the Instituto Politécnico Nacional in Mexico.

**Acknowledgments:** The authors wish to thank the Center of Nanosciences and Micro-Nano Technologies of the Instituto Politécnico Nacional for their cooperation.

**Conflicts of Interest:** The authors declare no conflict of interest. The funders had no role in the design of the study; in the collection, analyses, or interpretation of data; in the writing of the manuscript; or in the decision to publish the results.

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
