# Peer review of "Surface Modification of the Ti-6Al-4V Alloy by Anodic Oxidation and Its Effect on Osteoarticular Cell Proliferation"

_coatings, doi:10.3390/coatings10050491_

Round 1

Reviewer 1 Report

The article is interest to reader, but need some changes described below.

Remarks:

In 2.1. paragraph more details about used titanium alloy should be added: manufacturer, what kind of Ti-6Al-4V alloy was used (ELI? Grade 5?) and should be presented information about chemical composition according with manufacter certification.

In 2.1. paragraph more detail about used chemical reagents sholu be added (producer, concentration, purity).

3.2. paragraph:

In the authors' opinion, the tested Ti-6Al-4V surface before modification was hydrophobic for the measured contact angle 67.2. So what limits of hydrophobicity / hydrophilicity did the authors adopt and what reference did they use? According to the reviewer, all surfaces have a hydrophilic character, although of course it is different for the tested samples. In addition, the data presented on the chart are different from those presented in the figure and in the text.

The contact angle results data (Line 233-234) is also questionable. How many measurements have been made on each sample since standard deviations have integer values? In addition, for the value of 15.9 there is no standard deviation in the text, and all deviations are missing in the graph (Figure 7.h). The data submitted should be verified.

Editor and details remarks:

Line 31 – mistake “TiO2when”

Line 50 – after “chemical chomposition” there should be no comma

Line 239 – et all, should be “et al.”

Line 234 – (C2H6O2), should be C2H6O2

Author Response

Dear Reviewer,

We are very grateful for the time you spent on the revision of our manuscript.

The answers to your comments are described in the attachment document.

Thank you somuch

best regards

Enrique Hernández

Reviewer 2 Report

The manuscript reports the influence of anodic oxide layer formed on Ti6Al4V on biocompatibility and cell proliferation. Even if the topic could be interesting for the readers of Coatings I think that the manuscript is not well written for the following reasons.

Novelty:
It is not clearly indicated what is the novelty of this manuscript, i.e. designing the direct cell culture device or analysis the biocompatibility of titanium oxide nanotubes formed on Ti6Al4V (TNT). Furthermore, the manuscript claims that there are no complexity analysis of the cell-TNT interaction, most previous studies focusing on biocompatibility and antimicrobial properties [1-3].
Not only that such claims are debatable, as even the references cited in the introduction indicating "the 66 nm titanium inner nanotube diameter have a significant effect in the expression of adhesive molecules" do not correspond with the omitted papers [4-5]:

Methodology:
- The anodization process was performed under an applied voltage of 60V for only 60 min, why?
- How does the height of TNT change (evolve with anodization time)?
- The lack of error bars in figures 7 and in text (diameter and height of TNT).
- Furthermore, I think that all the data reported in Fig.4, display associated errors which are so big that it is difficult to make any consideration about the TNT diameter-anodic potential correlation.
- TNT were formed only in alpha phases of Ti6Al4V, while the modern techniques allow to form this layer on both phases [6].
- Fig. 5 - the TNT structure are non homogenous and destroyed, what influences the cell adhesion, and does not compare with current research.
- Cristallinity, it should be analysis which phases of Tio2 is dominant (anatase/rutile), what significantly influences the cell adhesion and proliferation.
- Fig. 7 - Why contact angle measured for ti6Al4V anodized for 30 min is so high?
- The discussion on hydrophilic surfaces for enhanced cell proliferation is very limited, especially the hydrophobic effect.

Conclusions:
I think that the novelty, relevance, and impact of the present manuscript are not very strong. It should be completed with the influence of varying diameter, height, anatase/rutile ratio of TNT on biocompatibility and what is more importantly, the nanotubular layer should be formed on both phases of Ti6Al4V.
Finally, the use of English needs to be thoroughly checked throughout the manuscript as the use of many words and the word sequence in numerous sentences are awkward.
Under these circumstances, in its present form, the manuscript is not fit for publication. In my opinion, a major revision is necessary before it can meet the requirements for publication in this reputable journal.

Completelly omitted papers (just a few examples):
[1]Filova, E., Fojt, J., Kryslova, M., Moravec, H., Joska, L., & Bacakova, L. (2015). The diameter of nanotubes formed on Ti-6Al-4V alloy controls the adhesion and differentiation of Saos-2 cells. International journal of nanomedicine, 10, 7145–7163. https://doi.org/10.2147/IJN.S87474
[2]Duvvuru, M. K., Han, W., Chowdhury, P. R., Vahabzadeh, S., Sciammarella, F., & Elsawa, S. F. (2019). Bone marrow stromal cells interaction with titanium; Effects of composition and surface modification. PloS one, 14(5), e0216087. https://doi.org/10.1371/journal.pone.0216087
[3]Luan, H., Wang, L., Ren, W. et al. The effect of pore size and porosity of Ti6Al4V scaffolds on MC3T3-E1 cells and tissue in rabbits. Sci. China Technol. Sci. 62, 1160–1168 (2019). https://doi.org/10.1007/s11431-018-9352-8
[4]Lan M-Y, Liu C-P, Huang H-H, Lee S-W (2013) Both Enhanced Biocompatibility and Antibacterial Activity in Ag-Decorated TiO2 Nanotubes. PLoS ONE 8(10): e75364. https://doi.org/10.1371/journal.pone.0075364
[5]Ahmad Barudin, N. H., Sreekantan, S., Thong, O. M., & Lay, L. K. (2012). Studies of Cell Growth on TiO2 Nanotubes. Advanced Materials Research, 620, 325–329. doi:10.4028/www.scientific.net/amr.620.325
[6]Arkusz, K.; Nycz, M.; Paradowska, E. Electrochemical Evaluation of the Compact and Nanotubular Oxide Layer Destruction under Ex Vivo Ti6Al4V ELI Transpedicular Screw Implantation. Materials 2020, 13, 176.

Author Response

(The authors gave the same response as above.)

Reviewer 3 Report

Introduction

“Additionally, electrochemical methods primarily anodic oxidation (AO) is not only a convenient but also a none expensive way to produce TiO2 nanotubes (TNT´s) [8], which had proved highly effective when used in tissue engineering. AO allows eroding the surface of metals through an electrochemical process by involving reactions between electrodes driven by an electric……tubular spaces have developed between the pores and the barrier layer on the metal-oxide interface [9].”

Please add more references, only one in a very long paragraph.

The authors must make more clear why their paper is new to the multiple studies that have been already published. 

“Materials and methods”

“The anodization process was carried out by using a 90 mixture of ethylene glycol (EG), 1 wt% of distilled water, and 0.5 wt% of NHâ‚„F.”

Please clarify the conductivity and pH of the electrolyte solution, did you use a slope to reach the desired voltage?

“fetal bovine serum (FBS)”

Please reference it with the appropriate batch.

hFOB cells must be maintained at 35ºC in order to avoid their differentiation, please explain why they were maintained at 37ºC.

“seeded with the SW-1353 cells (500 µL of a cell suspension of 105 cell/ml”

Please indicate the correct quantity, I assume it is not 105 cells.

“Cytotoxicity was measured by lactate dehydrogenase (LDH) releasing into the culture supernatant or present in attached lysed cells, as an indirect indicator of cell viability using the  CytoTox 96 non-Radioactive Cytotoxicity Assay kit (Promega, Wisconsin, USA)”

Please indicate the wavelength and the device the was used to measure it.

“The nature and the crystalline structure of the TNT´s were evaluated by XRD, as shown in Figure 6.”

The amorphous material was not analyzed? The F was not incorporated in the crystalline structure? Please identify if the TiO2 is anatase or rutile.

Did you expect any of the alloying elements (Al, V) to be incorporated into the nanotubes?

Why did the authors evaluate the different cell lines biocompatibility with two different techniques rather than the same for both?

Figure 8. A percentage of cytotoxicity or viability is advised with the LDH measurement or MTT and not the fold increase. Please redo the graphics.

Figure 3, 5 and 10- please improve the quality of the images, remove the microscope label information, and add a scale into the pictures.

Figure 6. Improve the quality of the graphics, they are very pixelated. Please consider introducing some colors to allow the audience to understand.

Figure 7. Improve the quality of the graphics, they are very pixelated.

Author Response

(The authors gave the same response as above.)

Round 2

Reviewer 2 Report

Authors made appropriate changes, my recommendation is to accept this paper for publication.

Author Response

Dear Reviewer,

We are very grateful for the time you spent on the revision of our manuscript.

Thank you very much.

Reviewer 3 Report

The manuscript has been improved, however, there are still some concerns about the following facts: 

"The structures of the TNTs achieved below 20 V were reported to be amorphous, while their crystalline structure was reported to appear at higher voltages. Moreover, the crystalline structure was reported to be anatase, rutile, or a mixture of both [26-28]. Thus, the higher potential used under these experimental conditions favors the formation of a crystalline structure (Figure 6)."

-Please, discuss and relate with other studies how the peaks change with treatment time, as with the longer treatment much less amorphous material is present and the main peak of TiO2 also differs with treatment time. 

-No F- was found with EDS either at 60 min nor in the crystalline structure? Please explain why, as in your own explanation suggest that it is incorporated:

"Finally, the ?− ions react with ??4+ and the other oxide species to form ???6
−2 (12), which is soluble, thereby facilitating the dissolution of the Ti cations: ???2 + 6?
− + 4?
+ → ???6−2 + 2?2? (11)
??4+ + 6?− → ???6-2 (12)"

-If V and Al are present on the surface, how the authors think that can affect cells after longer exposure times as it has been described as toxic? 

-Please relate the low contact angle result of the 60 min sample with the characterization results, it is due to the presence of less amorphous material or a higher formation of TiO2? Please discuss this fact. 

-"Figure 9. (a) SW1353 cell viability determined by the presence of the LDH enzyme"

It should be clarified that it is a total LDH assay not to be misinterpreted. 

"Figure 9" Add the cell culture moment that the assays were performed in the figure captions.

A different disposition of figures is advised, as it is confusing. 

Make one for the SW1353 cells and another one for hFOB cell. For a non-cellular culture audience it is hard to follow as you use different tests in different cell types with different samples at different culture days. You should discuss and detail why only the 10 and 60 min samples were evaluated for longer times. 

Explain why at 14 days where osteoblast should have to differentiate over the materials the metabolic activity is so high, this fact does not make sense. The authors should have evaluated mineralization, calcium deposits, ALP or other makers at 14 days. As it is known at the temperature of 33.5°C, the hFOB cells rapidly divide with a cell-doubling time of 36 h whereas at 39.5°C, cell divisions slowed with a doubling time of 96 h and it differentiated into mature osteoblasts. If cells are not dividing the metabolic activity cannot be high at such long exposure times. So based on your results, would the authors say that the material does not allow cells to differentiate? 

Relate the size of the cells to the size of the nanotubes and also the surface roughness, a very important parameter that has not been taken into account. 

Abstract:

"Moreover, in vitro assays using an innovative cell culture device demonstrates that the inner diameter of the nanotubes directly correlates with cell proliferation and adhesion."

The adhesion has not been evaluated and compared in all the samples, please rephrase.  

Author Response

Dear Reviewer,

We are very grateful for the time you spent on the revision of our manuscript.

Round 3

Reviewer 3 Report

The authors have greatly improved their manuscript and it can be published in the present form.